# Explainable Machine Learning Model for Glaucoma Diagnosis and Its Interpretation

**DOI:** 10.3390/diagnostics11030510

**Published:** 2021-03-13

**Authors:** Sejong Oh, Yuli Park, Kyong Jin Cho, Seong Jae Kim

**Affiliations:** 1Software Science, College of Software Convergence, Jukjeon Campus, Dankook University, Yongin 16890, Korea; sejongoh@dankook.ac.kr; 2Department of Ophthalmology, College of Medicine, Dankook University, 119, Dandae-ro, Dongnam-gu, Cheonan-si, Chungnam 31116, Korea; whitee11@naver.com (Y.P.); perfectcure@dankook.ac.kr (K.J.C.); 3Department of Ophthalmology, Institute of Health Sciences, Gyeongsang National University School of Medicine and Gyeongsang National University Hospital, Jinju 52727, Korea

**Keywords:** glaucoma, machine learning, prediction, model explanation

## Abstract

The aim is to develop a machine learning prediction model for the diagnosis of glaucoma and an explanation system for a specific prediction. Clinical data of the patients based on a visual field test, a retinal nerve fiber layer optical coherence tomography (RNFL OCT) test, a general examination including an intraocular pressure (IOP) measurement, and fundus photography were provided for the feature selection process. Five selected features (variables) were used to develop a machine learning prediction model. The support vector machine, C5.0, random forest, and XGboost algorithms were tested for the prediction model. The performance of the prediction models was tested with 10-fold cross-validation. Statistical charts, such as gauge, radar, and Shapley Additive Explanations (SHAP), were used to explain the prediction case. All four models achieved similarly high diagnostic performance, with accuracy values ranging from 0.903 to 0.947. The XGboost model is the best model with an accuracy of 0.947, sensitivity of 0.941, specificity of 0.950, and AUC of 0.945. Three statistical charts were established to explain the prediction based on the characteristics of the XGboost model. Higher diagnostic performance was achieved with the XGboost model. These three statistical charts can help us understand why the machine learning model produces a specific prediction result. This may be the first attempt to apply “explainable artificial intelligence” to eye disease diagnosis.

## 1. Introduction

Glaucoma is the leading cause of irreversible blindness worldwide and affects the optic nerve progressively [1]. It is diagnosed currently via four examinations: (1) detection of elevated intraocular pressure (IOP), (2) assessment of damage to the optic disc by calculating the cup-to-disc ratio (CDR), (3) identifying decreased retinal nerve fiber layer (RNFL) thickness, and (4) detection of characteristic visual field defects. Structural and functional clinical modalities such as optical coherence tomography (OCT) and visual field (VF) test provide indicators or values that can be used to diagnose glaucoma [2]. However, these conventional methods of glaucoma detection have higher chances of being misdiagnosed [2]. Hence, artificial intelligence (AI) systems are necessary to prevent misdiagnosis [3]. Machine learning has recently become one of the core technologies in the fields of science and technology, including life science and medicine. Classification is a major technology used in medical applications because it can be applied to prediction (diagnosis). The machine learning approach is also useful for predicting glaucoma. In the near future, machine learning will be an essential tool for predicting and treating glaucoma.

Despite the success of machine learning, machine learning has a fatal drawback for clinical applications. Most machine learning predictors do not explain the grounds of their individual prediction. As a result, they are known as a “black box.” In medical applications, this is a serious disadvantage. Medical personnel cannot trust a machine learning diagnosis without a reasonable explanation. We may trust a predictor based on a deep neural network model because it can show a high prediction accuracy against large test data. However, trusting individual prediction results is a different issue because the prediction may be incorrect. Even if the prediction is correct, it becomes difficult to trust without a reasonable basis. 

The Defense Advanced Research Projects Agency has recently initiated a new project called Explainable Artificial Intelligence, which shows that the explainability of AI has become a hot issue. Researchers have developed a large number of prediction models (predictors) based on machine learning. Most of them are mainly based on “deep learning” and use image data for prediction [4,5,6,7,8]. Traditional machine learning models have also been applied for glaucoma prediction [9,10,11]. Some papers suggested segmentation of areas with abnormalities on optical images [12,13,14,15]. However, interpretations of “individual prediction” for glaucoma diagnosis remain unexplored. Recently, interpretable machine learning has been gaining attention to explain “prediction models” and “individual prediction” [16,17]. 

The partial dependence plot (PDP) shows the marginal effect that one or two features (variables) have on the predicted outcome of a machine learning model [16]. A PDP can show whether the relationship between the target and a feature (variable) is linear, monotonic, or more complex. When features interact with each other in a prediction model, the prediction cannot be expressed as the sum of the feature effects because the effect of one feature depends on the value of the other feature [16]. H-statistic, variable interaction networks (VINs), and partial dependence-based feature interaction have been suggested to measure the feature interactions [18]. Feature importance in a prediction model can be measured in various ways. Local interpretable model-agnostic explanations (LIME) [19], Shapley Values [20], and SHapley Additive ExPlanations (SHAP) [21] have been suggested to explain individual predictions. Microsoft researchers published a unified framework for machine-learning interpretability [21]. They integrated previous research outcomes related to machine-learning interpretability into an open-source library called ‘interpretML.’ 

In this paper, we propose a machine learning model for glaucoma prediction with explanation functions for individual prediction. We mainly focus on explaining individual predictions. For this purpose, we attempt to build an “understandable prediction model” instead of a “highly accurate model”, based on the XGboost algorithm [22,23]. Five features from a visual field (VF) examination, RNFL OCT, and IOP test are used for glaucoma prediction. To explain individual predictions, gauge, radar, and SHAP charts are used. An explanation of glaucoma prediction provides a basis for the ophthalmologist to determine whether to trust the predicted results. Furthermore, ophthalmologists may obtain clinical insight from the explanation. Details are described in the Discussion section.

## 2. Materials and Methods

### 2.1. Participants

We collected the medical records of patients who underwent RNFL OCT and VF examinations at Gyeongsang National University Hospital between January 2012 and November 2020. To conduct the study, all patients underwent comprehensive ophthalmological examinations, which included slit-lamp bio-microscopy, best corrected visual acuity (BCVA), autorefraction (KR8800, Topcon, Tokyo, Japan), central corneal thickness (CCT) measurement (Pentacam, Oculus GmbH, Wetzlar, Germany), Goldmann applanation tonometry (Haag-Streit AG, Bern, Switzerland), a dilated fundus examination, and fundus and red-free fundus photography (Canon, Tokyo, Japan). An automated VF test was conducted using the 30–2 program Swedish interactive threshold algorithm standard on a Humphrey 740 visual field analyzer (Carl Zeiss Meditec Inc., Dublin, CA). Spectral-domain OCT (SD-OCT) images, obtained using the Spectralis^®^ (Heidelberg Engineering GmbH, Heidelberg, Germany) platform were used to measure the peripapillary RNFL (pRNFL) thickness. 

In total, 975 eyes (of 430 patients) with glaucoma (POAG (primary open angle glaucoma) or NTG) and 649 eyes (of 377 patients) without glaucoma were included. The inclusion criteria for normal eyes were a BCVA of 20/40, normal anterior segment on a slit-lamp examination, no RNFL defects in red-free fundus photographs, no visual field defects, and an intraocular pressure of ≤21 mmHg. The inclusion criteria for glaucomatous eyes were as follows: BCVA of 20/40 or better, a normal anterior segment on a slit-lamp examination, and diagnosis of glaucoma by three glaucoma specialists. The glaucoma diagnosis was based on characteristic glaucomatous structural changes to the optic disc accompanied by glaucomatous visual field defects. The criteria for a glaucomatous visual field defect were as follows: glaucoma hemifield test [24] outside the normal limit, pattern standard deviation with a *p* value of <5%, or a cluster of <3 points in the pattern deviation plot in a single hemifield (superior or inferior) with a *p* value of <5%, one of which must have a *p* value of <1%. Any of the preceding criteria, if repeatable, were considered sufficient evidence of a glaucomatous visual field defect.

In addition to those not meeting the inclusion criteria, the exclusion criteria were as follows: history of ocular inflammation or trauma, and the presence of concurrent retinal disease (i.e., vascular disorder or macular degeneration), optic nerve disease other than glaucoma, or a brain disorder that could influence the visual field results.

The inclusion criteria for normal eyes were a BCVA of 20/40, normal anterior segment on a slit-lamp examination, no RNFL defects in red-free fundus photographs, no visual field defects, and an intraocular pressure of ≤21 mmHg. Table 1 summarizes the characteristics of the participants.

### 2.2. Prediction Model

Figure 1 describes the prediction model development procedure. First, we collected clinical data of the patients based on a visual field test, an RNFL OCT test, general examination including an IOP test, and a Fundus image test. The first three are numerical data. In the case of fundus images, we build a prediction model based on a convolutional neural network (CNN), and the supported degree of glaucoma for each observation form feature data. As a result, our base dataset contains 22 features and a class label. To build an understandable model, we tried to choose a small number of features as quickly as possible. We tested 22 features, as shown in Table 2. 

First, we selected 10 informative features from the whole range of features using a chi-square feature selection measure. Then, a combination test was conducted to find a set of features that produces the best accuracy. Finally, we obtained five features that are described in Table 3.

Figure 2 shows a box plot of the five features. All features show a large difference in the median value between glaucoma and healthy controls. In particular, there is a big difference between Healthy control and glaucoma in pattern standard deviation (PSD). 

To build a prediction model for glaucoma, we tested various classification algorithms such as a support vector machine (SVM), random forest, C5.0, and XGboost. We use R packages to test the classification algorithms. To evaluate and compare the prediction models, we chose four criteria: receiver operating characteristic (ROC) plot, area under the curve (AUC), sensitivity, and specificity. The sensitivity of a prediction model refers to the ability of the model to correctly identify patients with a disease. The specificity of a prediction model refers to the ability of the test to correctly identify patients without a disease. The ROC plot expresses the relationship between sensitivity and 1 –specificity. The closer the ROC curve is to the upper-left hand corner, the better the model. AUC expresses the area under the ROC curve and can have any value between zero and 1. It is a widely used indicator of the goodness of a binary classification model.

From the comparison results, we confirm that XGboost shows the best performance with the hyperparameters described in Table 4. Therefore, we chose the XGboost prediction model for a glaucoma diagnosis. After model selection, we build a model explanation system, more exactly an explanation system for individual prediction results. The details are described in the next section.

### 2.3. Explanations of an Individual Prediction

If we input the values of the five features in Table 2 into the prediction model, either “glaucoma” or “healthy” is the output. To explain why the model produces the decision, we suggest three graphical charts, i.e., gauge, radar, and SHAP charts. The gauge and radar charts show the position of the input values among the complete distributions of the values. The SHAP chart shows the roles of each value in the decision.

Figure 3 (Left) shows an example of a gauge chart. In the dial, 0 and 170 indicate the minimum and maximum values of RNFL_S in a distribution of training data. In the dial, 64 and 155 are the boundary values of the overlapped range between glaucoma and healthy individuals in a distribution of the training data. In Figure 3 (Left), the value of 86 at the center of the bottom line indicates the value of RNFL_S for a patient, and determines the angle of the needle on the dial. Red and green zones in a dial indicate the range of glaucoma and healthy individuals, respectively. The yellow zone indicates the overlapped range of glaucoma and healthy individuals. 

A radar chart is a visualization method of multivariate data in the form of a two-dimensional chart of three or more quantitative variables represented on axes starting from the same point, as shown in Figure 4. We basically use a radar chart to express the distribution of five feature values for a patient who we want to predict. The second purpose of a radar chart is to observe the pattern of a polygon inside the chart. We can expect that the patterns of glaucoma and healthy patients will be different. Figure 4 is extracted from the average feature values of glaucoma and healthy patients. As we can see, glaucoma patients have higher PSD/IOP feature values and lower RNFL_ S/RNFL _ I/RNFL _T feature values than healthy patients. This is helpful for clinicians in making a glaucoma diagnosis and understand the prediction result of the proposed predictor.

An SHAP chart [21] is based on Shapley values [20]. In the Shapley values theory, a prediction can be explained by assuming that each feature value of the instance is a “player” in a game, where the prediction is the payout. Shapley values tell us how to fairly distribute the “payout” among the features (players) [16]. Figure 5 shows an example of a SHAP chart. In a SHAP cart, the Y-axis represents features and their values for the target patient. The X-axis in the SHAP chart represents the degree of support for glaucoma. Positive values (red bars) support a glaucoma diagnosis whereas negative values (green bars) support a healthy diagnosis. The longer the bar is, the stronger the support. The title of a SHAP chart represents the result of the prediction and its degree of certainty. In the case of Figure 5, the prediction result is “Glaucoma,” and its degree of certainty is 0.96 (96%).

In Figure 5, the PSD value of 10.51 strongly supports a glaucoma diagnosis, as does a RNFL_T value of 47. An RNFL_I value of 119 supports the diagnosis of a healthy individual. The RNFL_S and IOP values weakly support a glaucoma diagnosis. These indicators show why the model predicts the target patient as suffering from glaucoma.

## 3. Results

We divided 1624 cases into 80% training set and 20% test set. A total of 1306 cases are used for developing the prediction models, and 318 cases are used for evaluating the models. Four well-known classification algorithms are tested to build the prediction models, and xgboost shows the best accuracy. Table 5 describes a performance comparison of the four classification models. In the AUC, the proposed xgboost model has a 0.945 rate. The sensitivity of the proposed model is 0.941. This means that the proposed model accurately predicts glaucoma patients with an accuracy of 0.947. It also means that it shows a small false negative ratio. The specificity of the proposed model was 0.950, which shows a predictive power against the healthy controls.

In the machine learning prediction model, no features provide an equal contribution towards a prediction. Figure 6 describes the importance of the features in the proposed model. Here, importance refers to the contribution of each feature to the prediction task. In the proposed prediction model, RNFL_S, RNFL_I, and PSD have a stronger influence than RNFL_T and IOP.

In the prediction task, the roles of each feature are not independent, and they cooperate with each other. In other words, they contribute to the prediction task through an interaction. Figure 7 shows a feature interaction chart [25] for the five features of the proposed model. In the interaction table, the (i-th, j-th) cell shows the feature interaction between features Fi and Fj. The interaction value between IOP and RNFL_I was 0.0054. This indicates that their interaction increases the prediction accuracy by 0.0054. In other words, the combination of IOP and RNFL_I provides synergy to predict glaucoma. By contrast, IOP negatively interacts with RNFL_S and RNFL_T, which decrease the prediction accuracy by −0.0028 and −0.0003, respectively.

In this paper, we introduced a glaucoma prediction model and described its application. The use of our three explanation charts for a specific prediction are discussed in the next section.

## 4. Discussion

### 4.1. Diagnosis of Glaucoma

Glaucoma is an ocular disease that causes damage to the optic nerve because of the elevation of IOP and eventually a progressive visual field loss [26]. When ophthalmologists diagnose glaucoma, they measure the patient’s IOP and check the CDR through fundus examination. In addition, cpRNFL and/or mGCIPL thickness is measured using OCT, a standard modality for evaluating glaucomatous structural damage to the optic disc, and if there are some regions with reduced RNFL thickness on OCT, glaucoma is diagnosed by checking whether visual field abnormalities appear in the corresponding region in the visual field test, and by monitoring visual function in glaucoma [26]. Most of the glaucoma and AI-related articles searched are subject to screening and diagnosis of glaucoma [27]. Among them, there are many papers that diagnose glaucoma by detecting increased CDR or loss of neuro-retinal rim in the optic disc using fundus photography [28]. In addition, studies to diagnose glaucoma by extracting data from the OCT and VF tests have also been reported [29]. In recent times, machine learning models that use two or more of these diagnostic modalities in combination have been reported. Yohei et al. suggested that visual field abnormalities within the central 10° were predicted using SD-OCT in glaucoma patients, and Christopher et al. developed a deep learning system to identify glaucomatous visual field defects and predict the severity of VF defects using SD OCT [30,31].

Our study differs from previous reports because it maximizes the number of clinical values used to diagnose glaucoma. These include demographic and ocular factors that can potentially influence the diagnosis of glaucoma, such as age, IOP, spherical equivalent, axial length, and central corneal thickness. In addition, four values were obtained from the quadrant cpRNFL thickness map measured by OCT, and indices such as MD, PSD, VFI, and GHT were extracted in the VF test. Ultimately, this machine learning model was operated in the same situation that clinicians treat glaucoma patients, and it has improved to a level that ophthalmologists can refer to when they examine patients.

For clinicians to use it when actually examining and treating patients, the machine learning prediction model must have high reliability, that is, be highly accurate. In addition, the prediction model should provide users with clear information based on the results obtained. However, many machine learning models currently do not provide a clear rationale, and this is a barrier, i.e., that machine learning skills cannot be used in clinical settings. In fact, no evidence has been found in previous reports that have studied glaucoma and AI. To solve the “black box,” which is the fatal shortcoming of other machine learning models, our approach presented the basis of judgment with gauge, radar, and SHAP charts. By presenting the evidence of this judgment as glaucoma or healthy, our approach was able to move a little closer towards “explainable AI for glaucoma.”

### 4.2. Case Analysis of Prediction Results

We examined the results of our prediction model in several cases. Table 6 summarizes these cases. Details of the test results are described in the Appendix A. Cases 1 Appendix A are typical healthy and glaucoma patients, respectively, and our model accurately predicted them. All three charts consistently indicated healthy and glaucoma, respectively. On the other hand, in case 3 (Appendix A), CDR was increased, and RNFL defects were observed in the supero-temporal area. Therefore, we made a clinical diagnosis of pre-perimetric glaucoma, but the prediction model predicted it to be healthy. In case 4 (Appendix A), the CDR was healthy and the RNFL defect was not present. In a patient diagnosed as healthy, our prediction model predicted glaucoma. By reviewing the three charts, it can be observed that the decrease in the superior cpRNFL thickness has an effect on the diagnosis of glaucoma. This patient had high myopia of −11.0D, resulting in a decrease in cpRNFL thickness overall. 

Finally, case 5 (Appendix A) was diagnosed with glaucoma owing to increased CDR, RNFL defect, and VF defect in the corresponding area. However, the proposed model predicted it to be normal. When examining the rationale for deriving these results based on the three charts, it seems that normal pRNFL thickness and VF test influenced this. In fact, in optic disc shape and red-free fundus photography, this patient can be clinically diagnosed with superior segmental optic hypoplasia (SSOH) accompanied by early glaucoma. However, our model was determined to be normal based on RNFL thickness and VF test indices. A review of all patients showed that AI has difficulty diagnosing glaucoma, pre-perimetric glaucoma, early glaucoma (no reduction in RNFL thickness in OCT or no VF defect in VF test), high myopia, and it was a case of accompanying optic nerve abnormalities such as SSOH. In fact, glaucoma specialists often have difficulty diagnosing glaucoma in these patients immediately at their first visit. In this case, if we use the proposed approach to check the basis of judgment by referring to the SHAP chart, we will be able to receive help for the diagnosis of glaucoma.

### 4.3. Analysis of Missed Predicted Cases

The proposed prediction model produces 5.3% of missed predicted cases. Table 7 shows the confusion matrix of the proposed model. We tested the proposed model using all cases, and the results are described in Table 7. False positive was observed in 28 cases and false negative in 59 cases. We analyzed the mean of feature values against correct and missed predictions for glaucoma cases. In Table 8, missed prediction implies that the prediction is “healthy,” but the actual one is “glaucoma.” We can confirm that glaucoma cases with low PSD and high RNFL_S and RNFL_I values can easily be mispredicted as they are the characteristics of healthy cases. Table 9 shows that healthy cases with low RNFL_S and RNFL_I values can easily be mispredicted as glaucoma. We need to consider more information to avoid misprediction. This is a further research topic.

A feature interaction is a well-known phenomenon in machine learning prediction models. In a specific case of prediction, the features influence each other. In Figure 8, for example, two cases have the same PSD value of 1.39 and prediction results are “Healthy.” However, the degrees of support for Cases A and B are 0.84 and 0.93, respectively. In case B, RNFL_S and RNFL_I support “Glaucoma,” whereas they support “Healthy” in Case A; however, degree of support is increased from 0.84 to 0.93. This is because the degree of support of RNFL_T and IOP for “Healthy” is increased. The PSD may be more strongly influenced by RNFL_T and IOP than by RNFL_S and RNFL_I. Figure 9 shows the distributions of the degree of support according to the feature values. A wide distribution of such support for a feature value indicates a high influence from the other feature values. Five features show different non-linear distribution patterns. Furthermore, the distributions are different according to the range of values in a feature. There are points where the distribution changes rapidly in IOP and RNFL_T. We do not know the full meaning of these facts at this point, but we believe the facts imply an important hint toward a glaucoma diagnosis. This is a further topic of research.

## 5. Conclusions

We implemented our prediction model and three explanation charts on the web. The name of the system is Magellan (Appendix A). If users select a patient case, Magellan shows the prediction results and explanation charts. A standalone version of Magellan is also being developed. Ophthalmologists may refer to Magellan and obtain additional evidence before making a final decision. Magellan is particularly helpful when the results of IOP, fundus photography, OCT, and VF are not consistent. If the ophthalmologist uploads patient’s results to this system, it can help determine whether a patient has glaucoma. The diagnosis evidence provided by Magellan, an explainable AI system, improves reliability by giving clues for diagnosis to ophthalmologists who are not glaucoma specialists. By providing a basis for judgement when explaining to patients, Magellan will be also a useful tool for communicating with patients. 

## Figures and Tables

**Figure 1 diagnostics-11-00510-f001:**
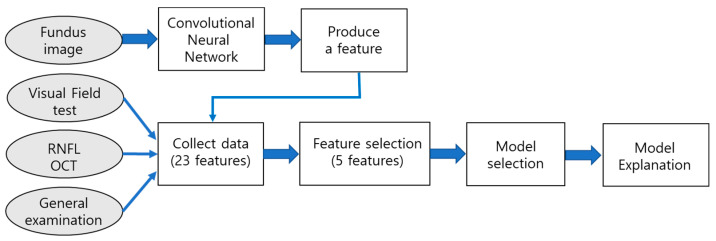
Procedure of prediction model development.

**Figure 2 diagnostics-11-00510-f002:**
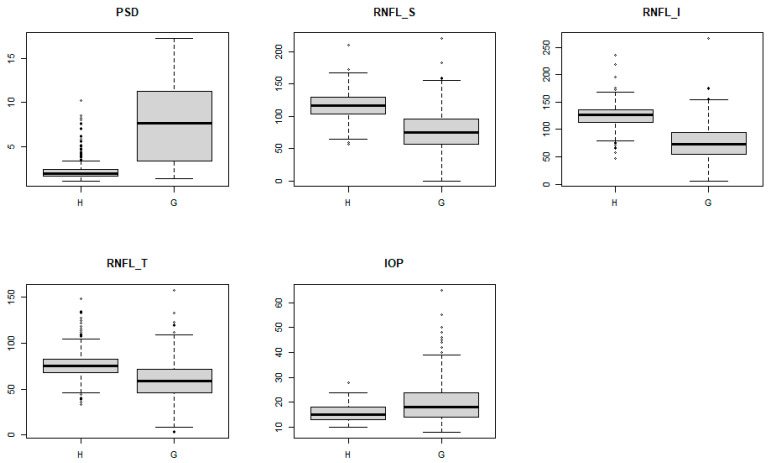
Box plots for selected features (H, healthy control; G, glaucoma).

**Figure 3 diagnostics-11-00510-f003:**
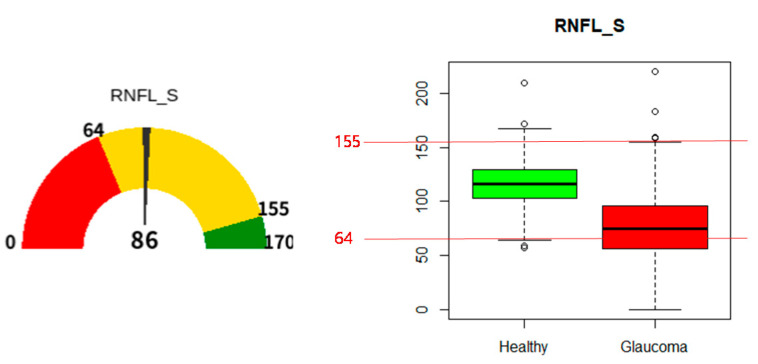
An example of a gauge chart and a boxplot for RNFL_T. (Left: gauge chart of RNFL_S with a value of 86, Right: boxplot and statistics of distributions for RNFL_S).

**Figure 4 diagnostics-11-00510-f004:**
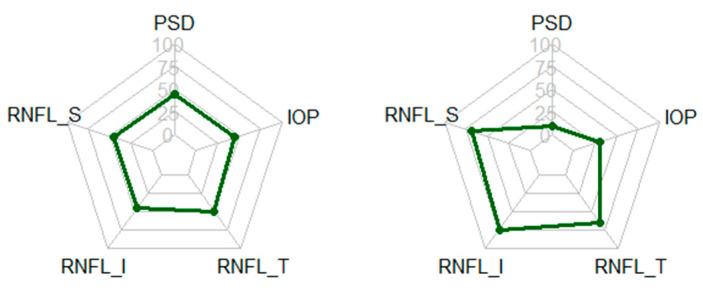
Radar charts for typical glaucoma and healthy patients.

**Figure 5 diagnostics-11-00510-f005:**
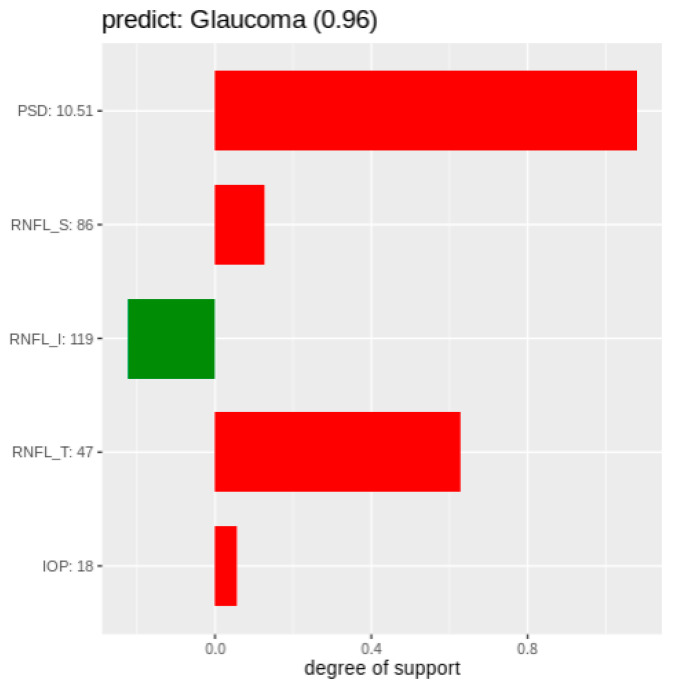
An example of a Shapley Additive Explanations (SHAP) chart.

**Figure 6 diagnostics-11-00510-f006:**
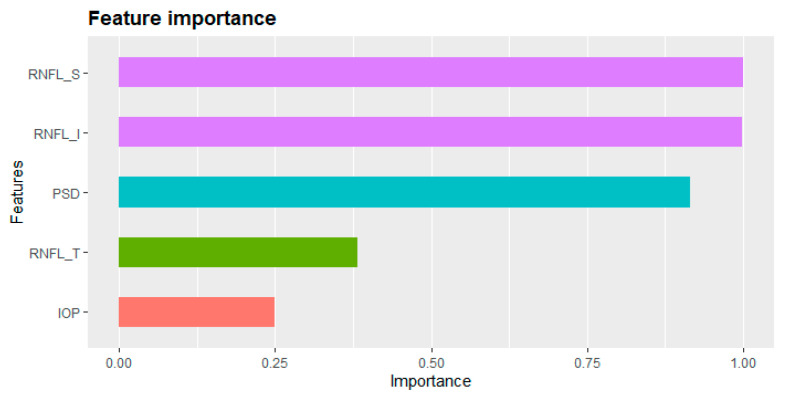
Feature importance of proposed model.

**Figure 7 diagnostics-11-00510-f007:**
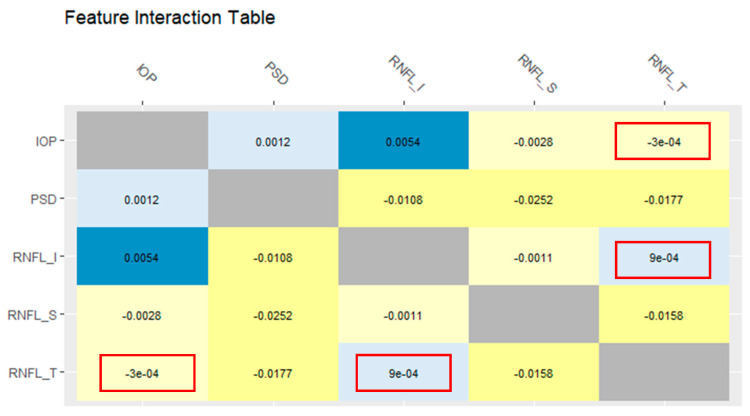
Feature interaction chart of the proposed model.

**Figure 8 diagnostics-11-00510-f008:**
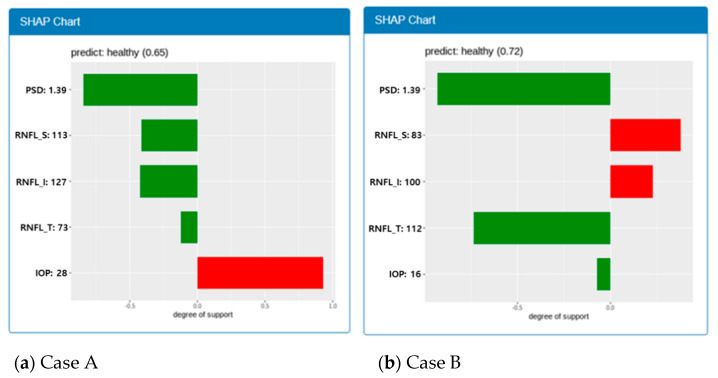
SHAP chart that shows feature interaction. The same PSD value of 1.39 has different weights in case A and case B.

**Figure 9 diagnostics-11-00510-f009:**
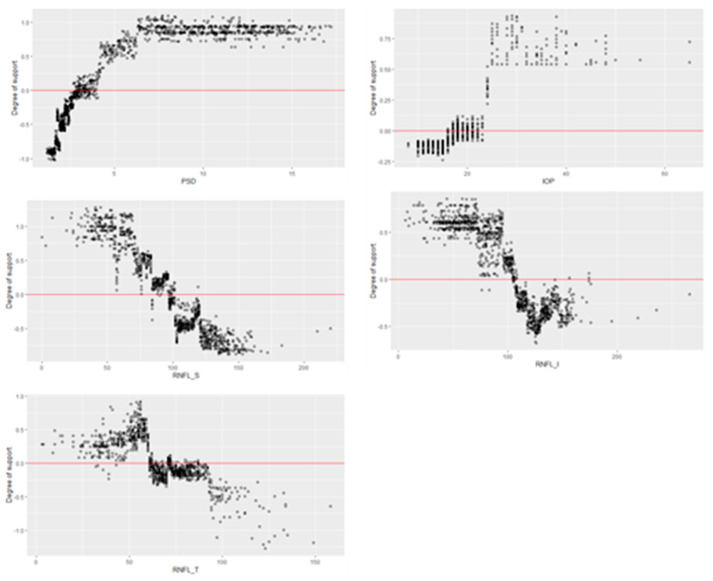
Distributions of degree of support according to the feature values.

**Table 1 diagnostics-11-00510-t001:** Characteristics of the participants.

Patient	Normal Group	Glaucoma Group	Total	*p*-Values *
Number of participants	377	430	807	-
Gender (male/female)	201/176	260/170	461/346	0.04061
Age (mean ± SD)	51.7 ± 16.5	60.3 ± 14.1	56.9 ± 15.7	<0.001
Number of eyes	564	680	1244	-
Number of cases	649	975	1624	-

* Normal group vs. Glaucoma group.

**Table 2 diagnostics-11-00510-t002:** List of candidate features for Model building.

No	Feature	Glaucoma Mean (SD ^1^)	Healthy Mean (SD)	*p*-Value
1	Sex	-	-	-
2	Age	60.3 (14.13)	51.7 (16.45)	<0.001
3	GHT ^2^	4.28 (1.28)	2.10 (1.53)	<0.001
4	VFI ^3^	72.3 (32.24)	95.7 (5.55)	<0.001
5	MD ^4^	−10.24 (9.72)	−2.33 (2.55)	<0.001
6	Pattern standard deviation	6.76 (4.27)	2.49 (1.03)	<0.001
7	RNFL ^5^ superior	82.0 (27.71)	112.6 (19.89)	<0.001
8	RNFL Nasal	56.1 (33.84)	64.6 (16.64)	<0.001
9	RNFL inferior	79.9 (29.76)	117.6 (20.28)	<0.001
10	RNFL temporal	58.6 (18.48)	71.6 (15.01)	<0.001
11	Mean of RNFL thickness	68.5 (18.82)	91.6 (12.60)	<0.001
12	Intraocular pressure	18.7 (8.69)	15.7 (3.10)	<0.001
13	Cornea thickness	527.2 (34.15)	530.1 (34.01)	<0.001
14	BCVA ^6^	0.63 (0.31)	0.73 (0.31)	0.002
15	Spherical equivalent	−1.63 (2.88)	−1.42 (3.08)	0.12
16	Axial length	24.1 (1.81)	24.1 (1.42)	0.92
17	Neuro-retinal rim	0.79 (0.28)	1.06 (0.21)	<0.001
18	Cup	0.47 (0.23)	0.38 (0.43)	0.16
19	Disc	1.97 (0.23)	2.09 (0.43)	0.25
20	Mean of cup/disc ratio	0.74 (0.11)	0.65 (0.12)	<0.001
21	vertical_cup/disc ratio	0.73 (0.10)	0.62 (0.16)	<0.001
22	CNN ^7^ degree	0.69 (0.18)	0.53 (0.21)	<0.001

^1^ standard deviation; ^2^ glaucoma hemifield test; ^3^ visual field index; ^4^ mean deviation; ^5^ retinal nerve fiber layer; ^6^ best-corrected visual acuity; ^7^ convolutional neural network.

**Table 3 diagnostics-11-00510-t003:** Final features list for building the prediction model.

No	Feature	Abbreviation	Source
1	pattern standard deviation	PSD	VF
2	RNFL superior	RNFL_S	RNFL optical coherence tomography (OCT)
3	RNFL inferior	RNFL_I	RNFL OCT
4	RNFL temporal	RNFL_T	RNFL OCT
5	intraocular pressure	IOP	IOP test

**Table 4 diagnostics-11-00510-t004:** Hyper parameters of proposed XGboost prediction model.

No	Hyper Parameter *	Value
1	booster	“gbtree”
2	eta	0.7
3	max_depth	8
4	gamma	3
5	subsample	0.8
6	colsample_bytree	0.5
7	objective	“multi:softprob”
8	eval_metric	“merror”
9	num_class	2

* We use default values for other hyper-parameters that are not listed in the table.

**Table 5 diagnostics-11-00510-t005:** Final features list for building the prediction model.

Metric	Support Vector Machine (SVM)	C50	Random Forest (RF)	xgboost
Accuracy	0.925	0.903	0.937	0.947
Sensitivity	0.933	0.874	0.924	0.941
Specificity	0.920	0.92	0.945	0.950
AUC	0.945	0.897	0.945	0.945

**Table 6 diagnostics-11-00510-t006:** Cases of prediction by the proposed model.

Case	PSD	RNFL_S	RNFL_I	RNFL_T	IOP	Diagnosis	Prediction
1	1.92	142	153	94	13	Healthy	Healthy
2	11.85	83	41	55	14	Glaucoma	Glaucoma
3	1.53	73	107	71	18	Glaucoma	Healthy
4	2.76	81	95	73	18	Healthy	Glaucoma
5	2.31	98	130	60	12	Glaucoma	Healthy

**Table 7 diagnostics-11-00510-t007:** Confusion matrix of the proposed prediction model.

		Predict	
		Glaucoma	Healthy
Actual	Glaucoma	916	59
	Healthy	28	621

**Table 8 diagnostics-11-00510-t008:** Analysis of prediction: glaucoma cases.

Case	PSD	RNFL_S	RNFL_I	RNFL_T	IOP
Correct prediction	7.99	75.31	72.86	57.98	20.73
Miss prediction	2.65	111.12	122.69	71.25	15.78
*p*-value	2.23 × 10^−55^	2.74 × 10^−24^	1.22 × 10^−34^	3.09 × 10^−10^	1.39 × 10^−17^

**Table 9 diagnostics-11-00510-t009:** Analysis of prediction: healthy cases.

Case	PSD	RNFL_S	RNFL_I	RNFL_T	IOP
Correct prediction	2.23	117.0	126.59	77.69	15.72
Miss prediction	3.84	90.40	91.21	61.89	15.68
*p*-value	1.64 × 10^−3^	1.74 × 10^−8^	2.66 × 10^−11^	1.27 × 10^−6^	9.44 × 10^−1^

## Data Availability

NA.

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
