# Peer review of "Explainable Machine Learning Model for Glaucoma Diagnosis and Its Interpretation"

_diagnostics, 2021, doi:10.3390/diagnostics11030510_

Round 1

Reviewer 1 Report

An innovative and five selected features (variables) were used to develop a 13
machine learning prediction model. The support vector machine, C5.0, random forest, and XGboost 14
algorithms were tested for the prediction model. The performance of the prediction models was 15
tested with 10-fold cross-validation. Statistical charts, such as gauge, radar, and SHAP, were used 16
to explain the prediction case. The proposed methodology is well and correctly described.

I have some remarks about the structure of the article. The following changes should be made:

  1. You should consider at least one paragaraph between table 2 and table 3.
  2. You should consider at least one paragaraph between table 3 and figure 2.
  3. Separate introduction section to 2 sections as 1. Introduction and 2. Background.
  4.  First and Second paragraph of introduction are too lengthy, split into 2-3 paragraphs.
  5. Good to see many literature reviews, but start with introduction.
  6. I have not seen conclusion in your paper. You can split your discussion to discussion and conclusion.

Author Response

#1. You should consider at least one paragraph between table 2 and table 3.
Answer: We added one paragraph table 2 and table 3.

#2. You should consider at least one paragraph between table 3 and figure 2.
Answer: We added one paragraph table 3 and figure 2.

#3. Separate introduction section to 2 sections as 1. Introduction and 2. Background.
Answer: Author guide line says that “The current state of the research field should be reviewed carefully and key publications cited” in the introduction section. In our opinion, background section is not general in this journal. 

#4. First and Second paragraph of introduction are too lengthy, split into 2-3 paragraphs.
Answer: We splitt second paragraph into three paragraphs.

#5. Good to see many literature reviews, but start with introduction.
Answer: This comment may be related with comment #3. 

#6. I have not seen conclusion in your paper. You can split your discussion to discussion and conclusion.
Answer: We split conclusion section according to your comment.

Reviewer 2 Report

The authors develop a machine learning model for the diagnosis of glaucoma and an explanation system for a specific prediction and they found that higher diagnostic performance was achieved with the XGboost model and three statistical charts can help us understand why the machine learning model produces a specific prediction result.

Major concerns

#1 Although the idea and topic seem OK, some sections must be condensed or resume to improve the comprehension.

#2 A practical approach to day life routine in ophthalmology work must be included in the last part of the discussion prior to conclusion.

#3 Avoid excess of information, some supplemental files could be removed.

#4 Include a small section on possible future lines of research that would be applicable if this technology is implemented.

#5 There is an excess of figures, so I advise you after reading my instructions below. Please check which of them would be essential and remove the ones that are not.

Minor concerns

#6 Line 12 Expand RNFL OCT at abstract (corrected along all abstract and manuscript abbreviation please)

#7 Line 31, number 2 is a reference or a number?

#8 Include a reference in Line 35.

#9 Line 40 provide a search strategy or delete the sentence.

#10 Line 56 prediction model section is confusing, rewrite.

#11 Line 91 What is the last C in BCVAC?

#12 Line 92 Goldman manufacturer and country.

#13 Line 99 patient number is missing.

#14 Line 114 Put inclusion criteria prior to exclusion one.

#15 Table 2 Express P value in a common way please

#16 Table 2 Expand abbreviation on legend.

#17 Figure 2 If you described first H and after G put in this order on the Figure.

#18 From Line 162 to 180, please resume and condensed these ideas.

#19 Structured the discussion into three main sections with the ideas very well differentiated.

#20 Figure 9 does not include valuable information, considered to removed or change the way you presented the information.

#21 Could be possible to improve visual perception of Figure 9. Quality is very poor, and the document have enough figures.

#22 Update references prior to 2010.

#23 Include only references from Journal Citation Reports.

#24 If possible, reduced volume of reference.

Author Response

Major concerns
#1 Although the idea and topic seem OK, some sections must be condensed or resume to improve the comprehension.
Answer: We condensed some sentences in discussion, and we split last paragraph into discussion section. Long paragraphs were split.  

#2 A practical approach to day life routine in ophthalmology work must be included in the last part of the discussion prior to conclusion.
Answer: As commented by the reviewer, the following has been added in the revised manuscript (Conclusion section).
“Magellan is particularly helpful when the results of IOP, fundus photography, OCT, and VF are not consistent. If the ophthalmologist uploads patient’s results to this system, it can help determine whether a patient has glaucoma. The diagnosis evidence provided Magellan, an explainable AI system, improves reliability by giving clues for diagnosis to ophthalmologists who are not glaucoma specialists. And by providing a basis for judgement when explaining to patients, Magellan is also a useful tool for communicating with patients.”
#3 Avoid excess of information, some supplemental files could be removed.
Answer: All supplemental files contains base information for discussion section that is one the ophthalmologist is interested in. In our opinion, they are They are needed to help understanding our paper.
#4 Include a small section on possible future lines of research that would be applicable if this technology is implemented.
Answer: We add conclusion section and describe further research. 
#5 There is an excess of figures, so I advise you after reading my instructions below. Please check which of them would be essential and remove the ones that are not.
Answer: We remove Figure 6 because it gives redundant information.

Minor concerns
#6 Line 12 Expand RNFL OCT at abstract (corrected along all abstract and manuscript abbreviation please)
Answer: We expended RNFL OCT and IOP at abstract. 

#7 Line 31, number 2 is a reference or a number?
Answer: It is a reference. We fixed it.

#8 Include a reference in Line 35.
Answer: We added a reference.

#9 Line 40 provide a search strategy or delete the sentence.
Answer: We removed the sentence.

#10 Line 56 prediction model section is confusing, rewrite.
Answer: We revised the sentence as follows:
Some papers suggested segmentation of areas with abnormalities on optical images.

#11 Line 91 What is the last C in BCVAC?
Answer: C is a typo. We removed it

#12 Line 92 Goldman manufacturer and country.
Answer: Goldmann applanation tonometry manufacturer and country name are described as follows. (Haag-Streit AG, Bern, Switzerland)

#13 Line 99 patient number is missing.
Answer: We revised the patient number.

#14 Line 114 Put inclusion criteria prior to exclusion one.
Answer: As suggested by the reviewer, we moved the description of inclusion criteria before exclusion criteria.

#15 Table 2 Express P value in a common way please
Answer: We revised it.

#16 Table 2 Expand abbreviation on legend.
Answer: We revised it

#17 Figure 2 If you described first H and after G put in this order on the Figure.
Answer: We revised it

#18 From Line 162 to 180, please resume and condensed these ideas.
Answer: We revised them to condensed paragraphs.

#19 Structured the discussion into three main sections with the ideas very well differentiated.
Answer: We revised the discussion into three sections as follows.
4.1. Diagnosis of glaucoma
4.2. Case analysis of prediction results
4.3. Analysis of missed predicted cases

#20 Figure 9 does not include valuable information, considered to removed or change the way you presented the information.
Answer: We revised the table caption.

#21 Could be possible to improve visual perception of Figure 9. Quality is very poor, and the document have enough figures.
Answer: We changed the figures.

#22 Update references prior to 2010.
Answer: We removed or updated references prior to 2010.

#23 Include only references from Journal Citation Reports.
Answer: We removed four references that are not from JCR.

#24 If possible, reduced volume of reference.
Answer: We removed less important references.

Round 2

Reviewer 2 Report

Comments solved